# Rubella Virus Infection, the Congenital Rubella Syndrome, and the Link to Autism

**DOI:** 10.3390/ijerph16193543

**Published:** 2019-09-22

**Authors:** Anthony R. Mawson, Ashley M. Croft

**Affiliations:** 1Department of Epidemiology and Biostatistics, School of Public Health, College of Health Sciences, Jackson State University, Jackson, MS 39213, USA; 2School of Pharmacy and Biomedical Sciences, University of Portsmouth, Portsmouth PO1 2DT, UK; ashley.croft@myport.ac.uk

**Keywords:** Rubella, infection, congenital rubella syndrome, CRS, autism spectrum disorder, vitamin A, retinoids, liver, pregnancy, vaccinations

## Abstract

Rubella is a systemic virus infection that is usually mild. It can, however, cause severe birth defects known as the congenital rubella syndrome (CRS) when infection occurs early in pregnancy. As many as 8%–13% of children with CRS developed autism during the rubella epidemic of the 1960s compared to the background rate of about 1 new case per 5000 children. Rubella infection and CRS are now rare in the U.S. and in Europe due to widespread vaccination. However, autism rates have risen dramatically in recent decades to about 3% of children today, with many cases appearing after a period of normal development (‘regressive autism’). Evidence is reviewed here suggesting that the signs and symptoms of rubella may be due to alterations in the hepatic metabolism of vitamin A (retinoids), precipitated by the acute phase of the infection. The infection causes mild liver dysfunction and the spillage of stored vitamin A compounds into the circulation, resulting in an endogenous form of hypervitaminosis A. Given that vitamin A is a known teratogen, it is suggested that rubella infection occurring in the early weeks of pregnancy causes CRS through maternal liver dysfunction and exposure of the developing fetus to excessive vitamin A. On this view, the multiple manifestations of CRS and associated autism represent endogenous forms of hypervitaminosis A. It is further proposed that regressive autism results primarily from post-natal influences of a liver-damaging nature and exposure to excess vitamin A, inducing CRS-like features as a function of vitamin A toxicity, but without the associated dysmorphogenesis. A number of environmental factors are discussed that may plausibly be candidates for this role, and suggestions are offered for testing the model. The model also suggests a number of measures that may be effective both in reducing the risk of fetal CRS in women who acquire rubella in their first trimester and in reversing or minimizing regressive autism among children in whom the diagnosis is suspected or confirmed.

## 1. Introduction

Rubella is a member of the togaviridae family, usually causing a benign systemic illness resembling a mild case of measles, and previously known as ‘germane’ (hence, ‘German’) measles [1]. The infection is characterized by rash, fever and lymphadenopathy. Up to 70% of infected adult women develop arthritis. If, however, infection occurs in the first weeks of pregnancy, up to 85% of neonates are born with a pattern of growth restriction and major birth defects known as congenital rubella syndrome (CRS). Later-onset sequelae of rubella in early pregnancy include autism and diabetes [2,3].

Rubella and CRS became nationally notifiable diseases in the United States in 1966. The highest number of notified rubella cases was in 1969, when 57,686 cases were reported at a rate of 58 new cases per 100,000 population. The incidence of rubella declined rapidly after vaccine licensure in 1969 and by 1983 fewer than 1000 cases per year were reported in the entire year (<0.5 cases per 100,000 population). A moderate resurgence occurred in 1990–1991 [4]. In the 12 months from December 2016 to November 2017, only 729 rubella cases were reported in Europe, of which 93 were laboratory-confirmed. The majority of cases (77.3%) were reported from Poland [1]. In the United Kingdom, the number of cases of acute rubella fell from 70 per year in 1988 to only 1 case in 1995 [5]. In poorer countries, rubella and CRS remain common. In the late 1990s, the incidence of CRS in these countries was estimated at between 44 and 275 cases per 100,000 live births [6].

Up to 60% of women of childbearing age worldwide remain susceptible to rubella and the CRS. In poorer countries, CRS is a major cause of developmental anomalies, especially blindness and deafness [7]. Worldwide, about 100,000 infants are born annually with CRS [8,9,10]. Fetal damage associated with rubella tends to result only when infection occurs in the first 16 weeks of gestation. In general, the earlier the onset of infection, the more severe are the observed malformations [11].

Although rubella infection and CRS are now rare in the U.S. and in Europe, due to widespread vaccination, autism rates have risen dramatically in recent decades to about 3% of children today, with many cases appearing after a period of normal development (‘regressive autism’). Here we review evidence suggesting that the signs and symptoms of rubella are due to infection-induced alterations in the hepatic metabolism of vitamin A (retinoids), causing mild liver dysfunction and the spillage of stored vitamin A compounds into the circulation in toxic concentrations, resulting in an endogenous form of hypervitaminosis A. Given that vitamin A is a known teratogen, it is proposed that initiation of the CRS in the early weeks of pregnancy is due to maternal liver dysfunction and exposure of the developing fetus to excessive vitamin A. On this hypothesis, the manifestations of CRS and associated autism represent endogenous forms of hypervitaminosis A. It is further suggested that regressive autism results primarily from post-natal influences of a liver-damaging nature and exposure to excess vitamin A, inducing CRS-like features as a function of vitamin A toxicity, but without the associated dysmorphogenesis. A number of environmental factors that may plausibly be candidates for this role are discussed below.

## 2. Maternal Rubella, Cataract, and the Congenital Rubella Syndrome

A major epidemic of rubella occurred in Australia in 1940, accompanied by reports of patients developing joint pain and arthritis. Until then, it was not known that rubella infection during pregnancy could adversely affect the fetus. Based on maternal reports, the Australian ophthalmologist Norman Gregg (1892–1966) was the first to describe an association between rubella infection in early pregnancy and congenital cataract. During the epidemic, Gregg saw 13 babies with bilateral congenital cataract and identified 78 additional cases through colleagues [12]. The cataracts involved all but the outermost layers of the lens, suggesting that cataract formation had begun early in gestation due to a common toxic or infective factor. Gregg’s hypothesis of rubella virus as the likely incriminated agent was subsequently confirmed [11], but the pathogenesis of CRS has remained obscure.

Among the affected infants seen by Gregg, many showed a failure to thrive along with feeding difficulties, both features suggesting heart defects. It has since been recognized that CRS includes ocular abnormalities, e.g., microphthalmia and retinopathy, as well as multiple systemic complications including the following [7,13,14]:bulging fontanelle and rash at birth;fetal growth restriction (FGR) and low birth weight;seizures;hearing and cardiovascular defects (most commonly patent ductus arteriosus);microcephaly;psychomotor retardation;behavioral and speech disorders;thrombocytopenic purpura;hepatitis;hepatosplenomegaly;bone lesions;pneumonitis;diabetes mellitus;thyroid disorders;progressive rubella panencephalitis

Discussing the significance of Gregg’s discovery, Webster noted that medical practice in the 1940s was still dominated by acute infectious diseases such as whooping cough, diphtheria, and meningitis [11]. Unlike most of the prevalent infections, which typically had high mortality rates, rubella in adults was considered a mild illness with coryza-like symptoms for 1–7 days followed by a 2- to 4-day rash, and then resolution of symptoms. There was therefore no reason to suspect that maternal rubella infection could severely damage the fetus, especially as birth defects were thought to be mostly genetic in origin.

## 3. Rubella Linked to Autism

Later research suggested that rubella infection in pregnancy was also linked to autism [15,16]. Many of the children were found to have an altered immune response to rubella vaccination, indicating congenital infection with rubella virus [17]. Stella Chess studied the behavioral features of preschool children with congenital rubella in New York City [18]. She identified the full syndrome of autism in 10 out of 243 such children and a partial syndrome in an additional eight (i.e., 18/243, or 7.4%). The diagnosis was based on Kanner’s criteria [19], notably a profound lack of interest in affective contact, coupled with repetitive and elaborate ritualistic behavior. This rate was over 200 times higher than estimated prevalence rates at the time, ranging from 0.7 per 10,000 in the U.S. to 2.1 per 10,000 children in London, England [20,21], and far higher also than a more recent reported autism rate in 2014 of 1 in 36 children among eight-year-olds in the U.S. [22]. Chess subsequently identified four additional cases in her cohort of children with CRS, increasing the autism rate to 10.24% [23].

Children with CRS in Texas in the 1960s were also reported to have a very high rate of autism. Of 64 children with CRS surviving at 18 months, 8 (12.5%) were diagnosed with the condition [15,16]. These data suggested a strong causal association between congenital rubella viral infection and autism. Later studies suggested links between other systemic viral infections and neurodegenerative conditions such as Alzheimer’s disease, Parkinson’s disease, amyotrophic lateral sclerosis, multiple sclerosis, and autism spectrum disorders [24,25,26].

Chess also observed a high rate of recovery from autism in children with CRS, hypothesizing that autism was secondary to chronic infection [18,23]. She noted that while many physical sequelae of rubella infection in utero—notably blindness, deafness, and cardiac and neuromuscular defects—remained unchanged in these children, the initial degree of mental retardation tended to diminish over time [27].

## 4. Congenital Rubella Syndrome—Case Study

In a case report that illustrates the condition [28], a 20-year-old Bangladeshi primigravid woman arrived in the U.K. when 20 weeks pregnant and was seen for prenatal care 2 weeks later. Fetal growth restriction (FGR) was noted on ultrasound scan. The baby was born at 31 weeks of gestation weighing 1035 g with features of symmetrical FGR as well as signs of intrauterurine infection, including jaundice, hepatitis, hepatosplenomegaly, thrombocytopenia, patent ductus arteriosus, and failure to thrive. At 6–8 weeks, early retinopathy and bilateral cataracts were noted on fundoscopy. Rubella virus IgM was noted 2 weeks later in serum samples. The mother recalled having a mild influenza-like illness without rash around the tenth week of pregnancy, confirmed by antibody avidity testing on paired maternal serum samples.

## 5. Rubella Infection and CRS—Introduction

Rubella infection is acquired by inhaling microorganisms exhaled, sneezed, or coughed by another infected individual. The virus infects cells in the upper respiratory tract, then spreads throughout the vascular system and replicates in lymphoid tissue of the nasopharynx, leading to the infection of multiple organ systems, including the placenta in cases of infection occurring early in pregnancy [7]. Following respiratory transmission and replication of virus in the nasopharynx and regional lymph nodes, viremia occurs 5–7 days after exposure, spreading throughout the body and transplacentally, leading to fetal damage via destruction of cells and mitotic arrest [9].

The neurologic sequelae of rubella infection can include a post-infectious encephalitis following acute infection, a spectrum of neurologic manifestations following congenital infection, and progressive rubella panencephalitis (PRP), a rare neurodegenerative disorder that can follow either congenital or postnatal infection. The pathogenesis of all three syndromes is thought to be autoimmune-mediated, as the virus has not been found in the brain following PRP [29]. Acquired and vaccination-associated rubella infection are also associated with post-infectious acute polyarthritis. In a series of 29 female patients who visited a general medical clinic in Japan, 33% had acute polyarthritis [30].

CRS-associated cardiovascular, central nervous system (CNS), hearing, and other systemic defects and growth restriction are attributed to direct damage to blood vessel walls and linings of the heart. Deafness, cardiovascular and neurological damage and retinopathy all appear when infection occurs in the first 16 weeks of gestation but are rare after that time [11]. The mechanisms underlying increased risk of CRS in the first trimester, followed by decreased risk in the second trimester, have not well understood but are suggested below.

Maternal rubella infection also results in a high frequency of *type 1 diabetes* in the offspring, typically presenting after many years [31]. Between 25% and 40% of young adults with a history of congenital rubella developed type 1 diabetes between ages 10 and 30, many having chronic thyroiditis or other autoimmune diseases [32]. Of 40 patients with congenital rubella born in 1939–1943 and seen again in 1967 and 1991, all had hearing impairments, eye defects (including microphthalmia) and vascular defects [33]. We now turn to the pathogenesis of rubella symptomatology, the congenital rubella syndrome, and links to autism.

### Retinoid Toxicity Hypothesis of Rubella Infection

It is proposed that the signs and symptoms of rubella infection are due to alterations in the hepatic metabolism of vitamin A, causing it to accumulate in and to damage the liver, and then to spill into the circulation in toxic concentrations. Although studies have not been reported on vitamin A in patients with rubella, there is evidence of liver involvement as well as strong parallels between the features of rubella and those of hypervitaminosis A. Polymorphisms in the vitamin A receptor and innate immunity genes have also been reported to influence the antibody response to rubella vaccine-induced immunity [34].

Our hypothesis is that rubella infection impairs the enzymes responsible for converting retinol to retinoic acid and for catabolizing retinoic acid, thereby increasing the concentration of vitamin A compounds (collectively termed retinoids) and causing liver damage and dysfunction. Serum retinol concentrations decline as a result of impaired hepatic mobilization and secretion of the carrier protein retinol-binding protein (RBP), while stored retinoid compounds from the dysfunctional liver enter the circulation, inducing the clinical features of rubella. If infection occurs in the early weeks of pregnancy, when fetal development is strongly regulated by endogenous retinoids [35], liver damage results in the entry of stored retinoid compounds into the fetal circulation in toxic concentrations, interrupting normal fetal development and causing dysmorphogenesis, brain damage and autism, among other systemic complications. We thus propose that the multiple manifestations of CRS, including autism, represent endogenous forms of hypervitaminosis A. It is further suggested that rising rates of autism and related neurodevelopmental disorders in recent decades result primarily from post-natal influences of a liver-damaging nature, also resulting in exposure to excess vitamin A and inducing CRS-like features as a function of vitamin A toxicity, but without the associated dysmorphogenesis.

## 6. Retinoids—A Summary

Vitamin A and its natural and synthetic congeners (‘retinoids’) are fat-soluble signaling molecules that are mainly derived from dietary intake or supplements and consumed either preformed from animals or else as beta-carotene (which is converted in the body to retinol) from fruits and vegetables. Retinoids are essential for multiple biological functions but can be highly toxic to cell membranes as well as mutagenic and teratogenic if unbound to protein [36,37]. Measurements of serum retinol are used to identify individuals with normal or deficient liver reserves, but they are homeostatically controlled by the liver and are neither closely correlated with vitamin A intake nor with signs of deficiency or excess [38,39].

Retinoic acid (RA), the main active metabolite, is produced from free retinol (ROL) via hydrolysis of retinyl esters (REs) in the liver. ROL is delivered to the target tissues bound to retinol-binding protein (RBP), then oxidized to retinal (retinaldehyde) via retinol dehydrogenase, and RA is synthesized from retinal through an aldehyde dehydrogenase reaction. RA controls the expression of over 500 genes through binding to and activating the nuclear protein receptors: retinoic acid receptors (RARs) and retinoid X receptors (rexinoids, RXR), both of which exist as three distinct gene products: alpha, beta, and gamma [40] (Figure 1).

Vitamin A is mostly stored in the liver (80%) but also in tissues such as the lung, adipose tissue, and intestine [41]. As noted, vitamin A can be very toxic if released into the circulation unbound to protein, i.e., as retinyl esters (REs) rather than as RBP. Retinoid toxicity can occur due to excessive dietary intake, supplements, and vitamin A medications. Vitamin A intake only marginally above the recommended amount is associated with embryopathy, reduced bone mineral density in the neonate, and increased risk for hip fracture [42]. REs destroy cell membranes and are a major source of vitamin A toxicity [43,44]. Retinoid toxicity can also occur in cholestatic liver disease due to the spillage of RA into the circulation in bile and the leakage of REs into the circulation from damaged hepatocytes [45]. Impaired hepatic mobilization lowers serum retinol, giving the appearance of a deficiency state but masking an overall state of retinoid toxicity. A variety of infections and stressors are associated with transient declines in plasma retinol concentrations [46,47], an effect that complicates the interpretation of serum retinol as an indicator of vitamin A nutritional status. Plasma retinol concentrations tend to rebound once the inflammatory stimulus is removed [48].

*Liver Dysfunction—*Case reports and reviews suggest that liver involvement is common in rubella infection and can include severe hepatitis, cholestasis, jaundice [49,50], and even fatal acute liver failure [51]. Liver dysfunction is often asymptomatic and up to 70% of patients with fatty liver do not show laboratory abnormalities, since liver function tests (which measure necrotic damage) provide only a crude indicator of abnormal liver function [52].

A 28-year-old male with acute hepatitis and acquired rubella infection showed marked increases in lactate dehydrogenase (LDH) activity and evidence of platelet and kidney injury, but only mild liver dysfunction [53]. Patients with autoimmune chronic active hepatitis and other chronic liver disorders can have very high titers of hemagglutination-inhibition antibodies to rubella virus [54]. Rubella infection is related to the hemophagocytic syndrome, a systemic clinico-pathological entity characterized by proliferation of benign hemophagocytic histiocytes, fever, cytopenia, abnormal liver function, coagulopathy and hepatosplenomegaly [55]. Neonatal hepatitis secondary to rubella occurs in about 10% of cases of infantile cholestasis [56].

## 7. Rubella Symptoms and Hypervitaminosis A

Consistent with the proposed model, there are strong parallels between the features of rubella infection and those of acute vitamin A intoxication (Table 1).

The symptoms and complications of postnatally acquired rubella infection include mild fever, arthralgia, polyarthritis, myalgia, headache, flu-like symptoms, conjunctivitis, lymphadenopathy, maculopapular rash, pruritus, fatigue, anorexia, and slight desquamation. Less common sequelae include encephalitis, splenomegaly, miscarriage, Guillain–Barré syndrome, and thrombocytopenic purpura [7].

Rubella is one of many viruses associated with arthritis and other musculoskeletal conditions [68]. An acute and generally mild nonerosive form of arthritis occurs in 10%–30% of infected persons, usually involving multiple small joints and lasting for 5–10 days post-infection, and most cases are short-term and self-limited [69]. However, rubella virus has not been recovered from peripheral blood lymphocytes in those with chronic arthropathy following rubella infection or vaccination [70]. On the present hypothesis, infection-associated acute and chronic joint pain and arthritis are due not to the virus itself but to the redistribution of vitamin A compounds from the liver to the affected joints in toxic concentrations.

Although usually a mild illness in children, the complications of rubella are occasionally severe. For instance, a 9-year-old boy with rubella encephalitis without rash was hospitalized with headache, fever, loss of consciousness and bilateral retroauricular lymphadenopathy [71]. Cerebrospinal fluid (CSF) examination revealed lymphocytic pleocytosis, increased protein levels and a normal glucose level. Immunoglobulin (Ig)M antibodies against rubella virus were positive in the CSF and serum. IgG antibody became positive in his serum 3 weeks after admission. Encephalitis similar to that seen with measles occurs in about 1 in 6000 cases. The clinical features of progressive rubella panencephalitis (PRP) resemble those of measles-associated subacute sclerosing panencephalitis (SSPE) but the age of onset of PRP is later and the clinical course is more benign. The main neurologic features are dementia, cerebellar ataxia, and seizures. Severity is highly variable, with a 20% mortality rate. Symptoms usually resolve within 1–3 weeks without neurologic sequelae. Increases in antirubella antibody titer and IgG are found in the CSF, associated with diffuse atrophy of the brain and ventricular dilatation [72].

*Acute vitamin A poisoning* is similarly associated with fever, arthralgia and arthritis, myalgia, headache, flu-like symptoms, conjunctival congestion, lymphadenopathy, pruritus, erythematous rash, weakness, anorexia, desquamation, and altered mental status. Less commonly but in more severe cases, hepatosplenomegaly, miscarriage, Guillain–Barré syndrome and thrombocytopenic purpura have been recorded [64,66]. Reports have been published of patients on oral isotretinoin treatment for acne experiencing thrombotic episodes resulting in cerebral ischemia, facial paralysis, and dysarthria [73].

Acute retinoid toxicity is associated with high-to-normal or even low circulating retinol concentrations, with increased fractions of retinyl esters circulating with plasma lipoproteins and not bound to RBP. Symptoms usually disappear after withdrawal of vitamin A, except for occasional liver enlargement [43,45,57,58,59,60,61,62,63,64,65,66,67]. The signs and symptoms of chronic vitamin A poisoning are discussed below in connection with CRS.

## 8. Congenital Rubella Syndrome (CRS)

The epidemic of rubella in Europe and the United States in 1964–1965 damaged thousands of children and showed that CRS has diverse manifestations affecting almost all organ systems. In addition to affecting the optic lens, cochlea and heart, the brain, lungs, liver, spleen, kidney, bone marrow, bones and endocrine organs are also affected in different degrees. The main defects result almost exclusively from rubella infection occurring in the first 16 weeks of gestation and include deafness, eye and cardiovascular defects, CNS damage leading to mental retardation, and a 50-fold increased risk of later-onset type 1 diabetes [31]. Associated features may include encephalitis, mental retardation, pneumonia, hepatitis, thrombocytopenia, metaphyseal defects, diabetes and thyroiditis [74].

Forrest et al. reevaluated 50 of Norman Gregg’s original patients with congenital rubella, born in 1939–1944 [75]. Echocardiography showed mild aortic valve sclerosis in 68%. Prevalence rates of diabetes (22%), thyroid disorders (19%), early menopause (73%) and osteoporosis (12.5%) were also increased compared with the general Australian population, and 41% had undetectable levels of rubella antibodies. Frequencies of HLA-A1 (44%) and HLA-B8 (34%) antigens were increased, while the haplotype HLA-A1-B8-DR3, which is strongly associated with autoimmune disease, was present in 25%.

## 9. Pathology of CRS

Based on a study of 57 therapeutic abortuses obtained after elective abortion in the first trimester of pregnancy [76], Webster noted the following [11]:

*Vascular defects*, including necrotic damage to endothelial cells in the capillaries and larger vessels of the placenta and myocardium; impaired development of the septum; heart malformations, especially patent ductus arteriosus (PDA) and pulmonary artery stenosis, associated with findings of generalized fibromuscular proliferation; and occasional occlusion of the arterial intima of large and medium-sized arteries. Related vascular abnormalities included focal destruction of the walls of cerebral blood vessels, defects of the internal elastic lamina with proliferation of fibrous tissue, and endothelial proliferation with narrowing of the lumen.

*Eye defects* included opacities of the primary lens fibers, resulting in a characteristic central or nuclear cataract. In neonates, the nuclear portion of the lens was often necrotic. A characteristic feature of rubella cataract was retention of nuclei in surviving lens fibers. The usually brief period of susceptibility to cataract was associated with the onset of maternal rash 12–43 days after fertilization. Damage to other ocular structures included focal necrosis of the pigment epithelium of the retina, necrosis of the ciliary body and iris, and microphthalmia. Compared to cataracts, the features of retinopathy and glaucoma occurred after infection over a much longer period (up to 117 days).

*Sensorineural deafness* associated with damage to the epithelium of the cochlear duct was the most common rubella-associated defect in surviving children.

*Brain damage* included mild to severe mental retardation associated with ischemia and variable microcephaly in about 25% of CRS patients. In vitro studies suggested that human astrocytes were selectively and heavily infected. MRI studies of CRS patients with ‘schizophrenia-like symptoms’ showed a reduced volume of cortical gray matter and significant ventricular enlargement. As noted, progressive rubella panencephalitis (PRP), a delayed manifestation of chronic infection in the brain, was associated with vasculitis, fibrinoid necrosis, severe neuronal loss and demyelination. Fetal growth restriction was a universal feature of CRS.

Many of the manifestations of CRS are present at birth but others are delayed, e.g., diabetes; thyroid disease; growth hormone deficiency; deafness; ocular damage (glaucoma, keratic precipitates, keratoconus, corneal hydrops, and absorption of the cataractous lens); vascular effects (including fibromuscular proliferation of the intima, arterial sclerosis, and hypertension, possibly secondary to renal disease, subretinal neovascularization), and PRP [77].

Theories of the pathogenesis of rubella-induced birth defects and delayed effects have included: viral proliferation in tissues, resulting in a reduced cellular growth rate and shortened life-span; autoimmune responses; genetic susceptibility; vascular damage from viral infection; reactive hypervascularization; and chronic persistence of the virus in the tissues [77]. Explanations have focused on the presence and access of the virus to the different structures, and on the timing of viral infection.

Of note, Nguyen et al. performed autopsies on aborted fetuses of mothers with a history of rubella with rash and fever at weeks 5–6 of gestation and found major histopathological changes in the liver, along with evidence of virus infection of the ciliary body of the eye, suggesting a possible cause of cataracts [78]. Lazar et al. carried out immunohistochemical staining of selected tissues of CRS fatalities and found rubella virus in interstitial fibroblasts in the heart and in adventitial fibroblasts of large vessels as well as in macrophages and progenitor cells of the outer layer of the brain [79].

## 10. CRS and Hypervitaminosis A-Associated Embryopathy

We propose that the morphology and behavioral effects of CRS are due to a single mechanism or process involving vitamin A toxicity, which influences multiple biologic functions and traits; namely, that rubella infection occurring early in pregnancy causes retinoids to accumulate in and damage the liver, leading to the spillage of toxic retinoid compounds (mainly retinyl esters and retinoic acids) into the circulation. Virus-induced changes in retinoid metabolism may thus be responsible for the malformations and damage to the fetal heart, blood vessels, brain and other organs involved in the early and late manifestations of CRS, as well as for stillbirth and rubella-associated autism. The pattern of outcomes may depend on the timing of exposure, the degree of hepatic involvement, and the concentration of retinoids. Long-term exposure of the tissues to retinoids may be responsible for the delayed manifestations of CRS. The model is summarized in the figure below (Figure 2).

Retinoids play a vital role in embryogenesis and can be highly teratogenic [35,80]. Retinoic acid, the biologically active metabolite of retinol, is a morphogen and a potent cause of numerous congenital defects, depending on the stage of gestation, dose and route of administration. Retinoic acid acts on the cell nucleus to change the pattern of gene activity by binding to specific ligand-activated retinoic acid receptors, which are members of the steroid-thyroid hormone-vitamin D superfamily. Retinaldehyde dehydrogenases (RALDHs) are the enzymes responsible for catalyzing the synthesis of retinoic acid from retinal, and for determining the tissue specificity of retinoic acid synthesis. RALDH2 in particular is required for the spatial and temporal concentration gradients of retinoic acid needed for proper embryonic development [81]. If consumed or administered early in pregnancy, retinoids can cause birth effects, fetal resorption and stillbirth in animals, and teratogenic sydromes in humans. Acute and chronic vitamin A supplementation of laboratory animals at therapeutic doses also impairs mitochondrial function in liver and brain, decreases brain-derived neurotrophic factor levels and dopamine D2 receptor levels, decreases glutamate uptake, and alters locomotor and exploratory activity [82,83]. Apoptosis has been suggested as the underlying and unifying mechanism of the adverse effects of retinoids in teratogenicity, depression, mucocutaneous side-effects, inflammatory bowel disease, myalgia, and release of transaminases due to liver injury. Genetic variants of components of the apoptotic signaling cascade may explain differences in susceptibility to the adverse effects of retinoids [84].

The retinoid toxicity hypothesis can explain why CRS manifests only when infection occurs early in pregnancy; it also accounts for the fact that the earlier the exposure, when retinoids are critically involved in morphogenesis, the greater is the extent of malformation. Owing to the prolonged elimination time of retinoids, effective contraception is recommended for at least two years following discontinuation of treatments with these drugs [85,86].

The effects of exogenous retinoids on the fetus include defects of the central nervous system (CNS) such as hydrocephalus and structural malformations of the cerebral cortex; cerebellar hypoplasia; craniofacial abnormalities; cleft palate; absent or reduced size of external ears; heart defects; reduced intellectual performance; limb defects; cataracts; microtia; maldevelopment of the facial bones; micrognathia; conotruncal heart malformation; and branchial arch defects [85,87]. In a report on 154 human pregnancies with fetal exposure to isotretinoin (13-cis-retinoic acid)—prescribed for severe nodular cystic acne but known to be teratogenic in laboratory animals—the relative risk for a group of major malformations was 25.6 (95% CI: 11.4, 57.5). The outcomes were 96 elective abortions, 26 infants without major malformations, 12 spontaneous abortions, and 21 malformed infants. Among the latter there was a characteristic pattern of malformation involving craniofacial, cardiac, thymic, and central nervous system structures. The malformations included microtia/anotia (15 infants), micrognathia (6), cleft palate (3), conotruncal heart defects and aortic-arch abnormalities (8), thymic defects (7), retinal or optic-nerve abnormalities (4), and central nervous system malformations (18). The pattern of malformation closely resembled that produced in animal studies of teratogenesis induced by retinoids [88].

The features of CRS closely resemble those of hypervitaminosis A-associated embryopathy and other chronic effects induced by therapeutic retinoids, hypervitaminotic diets, or supplements (see Table 2). As noted, the main features of CRS are eye defects (e.g., cataract, microphthalmia, and retinopathy); bulging fontanelle and intrauterine growth restriction; seizures; hearing and heart defects (including patent ductus arteriosus); microcephaly, psychomotor retardation, behavioral and speech disorders; hepatitis/jaundice and hepatosplenomegaly; bone lesions and osteoporosis; type 1 diabetes; thyroid disorders; and encephalitis. Each of these features has a counterpart in the effects of retinoid toxicity, as described below.

*Eye defects—*Cataract is a well-established feature of retinoid toxicity [85,87]. Both prenatal and postnatal exposure to isotretinoin are known causes of retinopathy and optic-nerve abnormalities [88]. Retinoic acid contributes to light-induced retinopathy in mice via mechanisms that may include plasma membrane permeability and mitochondrial poisoning, leading to caspase activation and mitochondria-associated cell death [94]. Mutations involving a loss of retinol dehydrogenase (RDH12) have been linked to severe retinal dystrophy involving light-induced retinal apoptosis in cone and rod photoreceptors. RDH12 acts as a retinaldehyde reductase, shifting the retinoid balance toward increased levels of retinol and decreased levels of bioactive retinoic acid. This activity protects against retinaldehyde-induced cell death and correlates with lower levels of retinoic acid in RDH12-expressing cells. Disease-associated mutants of RDH12 are unable to control retinoic acid levels in photoreceptor cells. RDH12 thus acts as a regulator of retinoic acid biosynthesis, protecting photoreceptors against overproduction of retinoic acid from all-trans-retinaldehyde [95].

*Bulging fontanelle and growth restriction—*Retinoid toxicity is a recognized cause of bulging fontanelle and growth restriction in infants [91,96]. Abrupt arrest of embryonic growth is also a known effect of retinoic acid and a characteristic feature of hypervitaminosis A [57,91,97]. Growth arrest has also been observed in rats fed retinoic acid [98].

*Seizures—*Chronic consumption of large amounts of vitamin A is associated with seizures in animals [99]. Vitamin A supplementation or therapy using synthetic retinoids are also associated with seizures [100,101]. Interestingly, however, direct infusion of retinoic acid into the brain can reduce seizure activity. Noting that the basolateral amygdala (BLA) plays an important role in the induction and control of seizures, Sayyah et al. reported that in amygdala-stimulated rats, the effect of intra-BLA infusion of retinoic acid had a rapid anti-seizure effect [102]. Intra-BLA infusion of retinoic acid also prevented the proconvulsant effect of trimethylamine, a drug known to enhance seizure susceptibility. The anti-seizure effect of exogenous retinoic acid may be due to feedback inhibition of retinoic acid synthesis, since brief isotretinoin treatment is known to lower serum retinol levels [103,104].

*Hearing loss—*Retinoic acid is necessary for normal development of the mammalian cochlea (the organ of Corti), which contains sensory hair cells that are essential for hearing [105]. However, sensorineural hearing loss is documented in many case reports as a complication of treatment with synthetic retinoids. For instance, an infant who had been exposed to 10 mg/day of acitretin, an aromatic retinoid analog of vitamin A, from the beginning of pregnancy until the 10th gestational week showed bilateral sensorineural deafness at term [106]. Serious ear malformations or absent external ears and/or auditory canals in infants were reported in mothers consuming 25,000 IU or more of vitamin A per day during pregnancy or were exposed prenatally to isotretinoin. Such exposures interfere with cranial neural crest cells, which contribute to development of the ear [88,107,108].

*Heart defects—*Cranial neural crest cells also contribute to the conotruncal area of the heart. Prenatal exposure to isotretinoin is likewise associated with conotruncal heart defects [88]. Several studies have reported associations between high vitamin A dietary and/or supplement intakes and cardiac defects [89]. The results have been somewhat inconsistent, possibly due to differences in methods of assessing exposure to vitamin A intake. Two studies have reported increased risks of congenital heart defects associated with supplement intakes greater than or equal to 10,000 IU retinol [109,110]. The use of isotretinoin in pregnancy has been specifically linked to patent ductus arteriosus [90].

*Microcephaly—*Small head circumference is a common malformation of the central nervous system associated with hypervitaminosis A in early pregnancy [106,111,112]. Benke described two infants with microcephaly, prominent frontal bossing, hydrocephalus, microphthalmia, and small, malformed, low-set, undifferentiated ears whose mothers had taken isotretinoin in the first trimester of pregnancy [113]. A Dandy-Walker malformation (involving the cerebellum and fourth ventricle), hypertelorism, small ear canals, cleft palate, small mouth, and congenital heart disease were also observed.

The same newborn infant exposed to 10 mg/day of acitretin (mentioned above under *Hearing Loss*) from the beginning of pregnancy until the 10th gestational week showed the following additional abnormalities: microcephaly, epicanthal folds, low nasal bridge, high palate, cup-shaped ears, anteverted nostrils, and atrial septal defect. At 18 months of age, the patient showed neurodevelopmental delay [106]. The role of retinoic acid metabolism in normal brain development and in embryopathy has also been proposed to explain reported cases of Zika virus-associated microcephaly in newborns in Brazil [114,115,116].

*Cognitive deficits, behavioral and speech disorders—*Although less well described and understood than the physical effects, cognitive, behavioral and speech disorders have also been documented from exposure to excess retinoids. These features are psychomotor retardation, a generalized slowing of mental and physical activity and skills, impaired attention, learning and speech disorders, emotional instability, depression and aggression [100,117,118,119]. Retinoid receptors are abundant in the brain structures involved in behavioral and cognitive functions, notably the striatum and hippocampus. In mice and rats, using maze running to assess memory functions, chronic exposure to a clinical dose (1 mg/kg/d) of 13-cis-RA suppressed hippocampal neurogenesis, disrupted hippocampal-dependent memory and impaired spatial learning [120,121,122]. In rats, chronic oral treatment with isotretinoin resulted in enduring memory deficits at doses that induce blood levels comparable to those used to treat acne in humans [123]. However, another study found that chronic 13-cis-RA treatment in male and female rats had little impact on measures of spatial learning and memory [124]. These inconsistent results may be due to differences in age, species, dose and route of administration [118], but many patients taking 13-cis-RA show signs of behavioral disinhibition and affective lability [119].

*Hepatitis/jaundice, hepatosplenomegaly—*Liver damage from prolonged excess dietary intake of vitamin A, supplementation or treatment with retinoids is a well-established feature of retinoid toxicity. High concentrations of vitamin A are toxic to the liver and spleen, producing hepatosplenomegaly and eventually cirrhosis [43,92,104,125,126]. Vitamin A hepatotoxicity is associated histologically with the accumulation of perisinusoidal lipocytes, associated fibrosis, obstruction of sinusoids and terminal venules, sclerosis of central veins, atrophy of adjacent hepatocytes, and an increase in basement membrane-like material and collagen within the perisinusoidal space in association with lipid-filled Ito cells [127,128].

*Bone lesions, osteoporosis*—Consistent with reports of increased risks of osteoporosis in association with CRS [75], chronic and excessive vitamin A intake causes inhibition of bone growth, progressive ligament calcification, modeling abnormalities of long bones, metaphyseal irregularities and osteoporosis [60,129]. Bone pain, refusal to walk, and fatigue are well-documented symptoms of hypervitaminosis A [92,130].

*Type 1 diabetes*—A 50-fold increase in the long-term risk of type 1 diabetes was reported in children with CRS [31]. Indeed, retinoids are strongly implicated in diabetes [41,93,131]. Here it is suggested that that exposure of the fetus and infant to toxic circulating concentrations of retinoids resulting from liver damage in turn damage the insulin-producing cells of the pancreas, thereby increasing the risk of type 1 diabetes in early life.

*Thyroiditis—*Thyroid disorders in patients with congenital rubella were first reported in 1975 [132]. Infection of thyroid tissue by rubella was demonstrated in a case of congenital rubella with Hashimoto’s thyroiditis (hypothyroidism) [133]. Thyroiditis encompasses a heterogeneous group of disorders characterized by thyroid inflammation. Autoimmune thyroid disease includes Hashimoto’s thyroiditis, characterized by lymphocytic infiltration and the presence of serum anti-thyroperoxidase antibody (TPOAb) and/or anti-thyroglobulin antibody (TgAb). In the follow-up study of Gregg’s original patients, 60 years after their intrauterine infection, the prevalence of thyroid disorders (and of diabetes and menopause) was higher in patients with congenital rubella than in the general population, yet 41% of the subjects had undetectable levels of rubella antibodies [75]. Most reports have shown no evidence of active rubella infection at the time of thyroid dysfunction. The mechanisms suggested for thyroid dysfunction have included destruction of thyroid cells by local persistent rubella virus infection and/or precipitation of an autoimmune reaction [134]. We have argued that drug-induced hepatocellular damage causes, firstly, a spillage of retinoids into the circulation (and hence a direct toxic effect on the brain and other target organs); and secondly, disruption of the liver-thyroid axis and hence a pattern of specific bipolar symptoms such as is often seen in thyroid disease [135]. Rubella-induced hepatocellular damage may likewise disrupt the liver-thyroid axis, giving rise to the thyroid sequelae of rubella that have been commonly described in the literature. Consistent with the retinoid toxicity hypothesis, pharmacologic doses of vitamin A adversely affect multiple aspects of thyroid function, e.g., decreasing total T4 and T3 levels, possibly by increasing peripheral clearance of thyroxine, but without altering basal thyroid stimulating hormone (TSH) or its response to thyroid-releasing hormone. Vitamin A decreases tissue responsiveness to thyroid hormones and decreases thyroid gland size; it also modulates thyroid gland metabolism, peripheral metabolism of thyroid hormone and production of thyrotropin (TSH) by the pituitary [136].

*Encephalitis/Encephalopathy—*The main neurologic features of progressive rubella panencephalitis (PRP) are dementia, cerebellar ataxia, and seizures. Severity is highly variable, with an overall mortality rate of 20%. PRP is thought to be due to viral infection of the central nervous system. In children with congenital rubella infection the neurologic deficits tend to remain stable, and deterioration after the first few years of life is rare. In some cases of congenital rubella, progressive neurologic illness can develop in the second decade, characterized by spasticity, ataxia, intellectual deterioration and seizures. In three such cases, high antibody titers to rubella virus in serum and spinal fluid were present in two cases, and all three had increased CSF protein and gamma globulin. Attempts to recover a virus from brain and body fluids were unsuccessful. The brains of two patients showed a widespread and progressive subacute panencephalitis, mainly affecting white matter [137].

Several indirect observations support a role for retinoid toxicity in the pathogenesis of PRP. First, chronic vitamin A poisoning is a known cause of both liver failure and encephalopathy. In one such case, a 60-year-old male with symptoms of muscle soreness, alopecia, nail dystrophy and ascites continued to deteriorate and developed refractory ascites, renal insufficiency, encephalopathy, and failure to thrive. Liver biopsy demonstrated the presence of fat-laden Ito cells and vacuolated Kupffer cells without the presence of cirrhosis. The clinical history revealed ingestion of large doses of vitamin A [138]. Second, vitamin A intoxication creates a syndrome that includes pseudotumor cerebri, psychosis, papilledema, and acute encephalitis [139,140]. Third, in the related condition of measles encephalopathy, serum retinol concentrations are low and are assumed to indicate vitamin A deficiency. For instance, among 21 patients with newly diagnosed subacute sclerosing panencephalitis and 20 age-matched controls, serum retinol <20 microg/dL was observed in 6 of the SSPE patients but in none of the controls (*p* < 0.05) [141]. As noted, however, low serum vitamin A concentrations can imply a state of impaired hepatic mobilization and secretion of vitamin A, with the associated possibility of the entry of unbound retinyl esters and retinoic acids directly into the circulation, resulting in vitamin A toxicity. On this hypothesis, retinoids pass the blood–brain barrier and likewise induce the symptoms and signs of encephalopathy.

## 11. Preventing CRS

The proven primary prevention strategy for CRS is, and will continue to be, the vaccination of young women [1]. Some women, mainly those who were not vaccinated, will continue to acquire rubella infection in the first trimester of pregnancy, so their unborn fetus will be at risk of CRS. In many countries, this is an indication for elective abortion. For those who do not wish to take this step, the following simple measures—proposed by us earlier in a different clinical context [135]—could be taken by the mother to reduce the expression of excess retinoids and hence lower the risk of fetal CRS.
stopping all alcohol for a period of at least 4 weeks;stopping all liver-damaging drugs that are not essential to life (e.g., anabolic steroids, recreational drugs, antidepressants, anxiolytics and hypnotics);maintaining good hydration by drinking plain, non-fluoridated water (and not through drinking coffee or tea, both of which are dehydrating);eliminating vitamin A from the diet for one month (the main dietary source is liver, but vitamin A is also found in milk, cheese, egg yolks and fish oils);(optional) reducing circulating retinoids, under medical supervision, through phlebotomy, plasmapheresis or hirudotherapy.

## 12. Retinoid Hypothesis of Regressive Autism

As noted above, from 7% to 12% of children with CRS were diagnosed with autism in the 1960s, rates that were 200-fold or higher than population-based rates at the time. Rubella and CRS are now rare in the U.S. However, rates of autism have increased dramatically, from about 1 in 5000 in the 1960s to about 3% today, with autism diagnoses having tripled and accelerated since 2000 [22].

Why has autism become epidemic in the U.S., and how does it relate to rubella and CRS? Many signs and symptoms are common to both CRS and autism, such as hearing deficits and auditory hypersensitivity. Hutton has suggested that CRS and autism are part of a spectrum of disease and that maternal rubella infection could still cause autism, even in vaccinated populations ([142], p. 1). On the other hand, from 30% to 40% or more children with ASD today have a regressive form of autism [143] in which an initial phase of normal development is followed between ages 1 and 2 years by the loss of acquired language skills and speech, and withdrawal of interest in the social environment [144].

We have suggested that the strong association between CRS and autism during earlier epidemics of rubella occurred via rubella virus infection-induced alterations in vitamin A metabolism in the early weeks of pregnancy, causing maternal liver damage and the entry of stored retinoids into the fetal circulation in toxic concentrations. Here it is proposed that rising rates of autism are due primarily to post-natal factors, similarly causing liver damage and endogenous forms of vitamin A intoxication.

One environmental factor to which virtually all children are exposed as a requirement for school and daycare attendance is the receipt of multiple vaccinations and their components in close succession as part of the *routine childhood vaccination program*. While the benefits of vaccines are well documented [145,146], few vaccines or vaccine combinations or their ingredients have been subjected to randomized controlled trials, observational studies or other types of tests for long-term safety. The measles, mumps and rubella (MMR) vaccine is one of few vaccines studied to date in relation to autism [147,148,149] 

Concerns that individual vaccines and/or the vaccination schedule itself could be contributing to increasing rates of autism [150,151,152,153,154] include the following:The vaccination schedule has been expanded and accelerated during the same period of rapid increase in rates of autism and other chronic illnesses. The schedule now involves 69 doses of 16 vaccines between the day of birth and age 18, with 50 doses before age six [155], which is over three times the number of vaccinations recommended in 1983 [156]. The clinical outcomes of this expanded schedule on children’s health, where many vaccinations are administered at the same time or in close succession, are virtually unknown [157].The vaccine adjuvant aluminum is a known neurotoxin [158] and has been implicated in autism and other neurodegenerative diseases [159,160,161,162,163].Vaccines are known on occasion to cause severe adverse effects, including immune system disorders, brain damage and death [164]. These outcomes are considered so rare that vaccines are believed safe to administer to all healthy infants and children [165,166]. However, the extent of serious injury from vaccines remains uncertain. One study of infants who were hospitalized or had died after receiving vaccinations, based on 38,801 reports to the Vaccine Adverse Events Reporting System (VAERS), showed a linear relationship between the number of vaccine doses administered at one time and the rate of hospitalization and death; furthermore, the younger the infant at the time of vaccination, the higher was the rate of hospitalization and death. The rate of hospitalization for two vaccine doses was 11% and increased to 23.5% for eight doses (r^2^ = 0.91). The case fatality rate increased from 3.6% for 1–4 doses, increasing to 5.4% among those receiving 5–8 doses [167].

No officially sponsored study has compared autism rates in vaccinated and completely unvaccinated groups of children, as recommended by the Institute of Medicine [168]. To address this gap in knowledge, the first author and colleagues collaborated with the National Home Education Research Institute, Salem, Oregon, to compare vaccinated and unvaccinated homeschooled children on a broad range of health outcomes based on mothers’ anonymous reports [169]. Four states were selected for the online survey: Florida, Louisiana, Mississippi and Oregon. Mothers were asked to provide information on their 6- to 12-year-old biological children. The requested information included pregnancy-related conditions and exposures, birth history, vaccinations, physician-diagnosed illnesses, medications, and the use of health services. The survey yielded fully completed questionnaires on 666 children, among whom 261 (39%) were unvaccinated. As expected, the vaccinated were significantly less likely than the unvaccinated to have been diagnosed with chickenpox and whooping cough. However, the vaccinated were significantly more likely to have been diagnosed with the following conditions: allergic rhinitis (Odds Ratio 30.1; 95% Confidence Interval: 4.1, 219.3); eczema (OR 2.9; 95% CI: 1.4, 6.1); middle ear infection (OR 3.8; 95% CI: 2.1, 6.6); pneumonia (OR 5.9; 95% CI: 1.8, 19.7); Attention Deficit Hyperactivity Disorder (OR 4.2; 95% CI: 1.2, 14.5); Autism Spectrum Disorder (OR 4.2; 95% CI: 1.2, 14.5); and a learning disability (OR 5.23; 95% CI: 1.6, 17.4).

The vaccinated were also more likely to use medication for allergies (OR 21.5; 95% CI: 6.7, 68.9), to have used antibiotics in the past 12 months (OR 2.4; 95% CI: 1.6, 3.6), been fitted with ventilation ear tubes (3.0% vs. 0.4%, *p* = 0.018; OR 8.0; 95% CI: 1.0, 66.1), and to have spent one or more nights in a hospital (19.8% vs. 12.3%, *p* = 0.012; OR 1.8; 95% CI: 1.1, 2.7).

Children who had received some but not all recommended vaccinations (i.e., were partially vaccinated) had significantly increased but intermediate odds of allergic rhinitis, eczema, attention deficit hyperactivity disorder (ADHD), a learning disability, and any chronic disease between those of unvaccinated and fully vaccinated children, suggesting a dose-response relationship between vaccinations and adverse effects.

Neurodevelopmental disorders as a group (defined as combining ASD, ADHD and learning disorders) were also associated with male gender and with preterm birth, as would be expected from the literature. The effect of preterm birth on neurodevelopmental disorders has not been studied separately from vaccinations, which are routinely administered to prevent infections. In a final adjusted statistical model with interaction, in which preterm birth and vaccination were studied separately and combined, vaccination remained associated with NDD; but preterm birth itself, without vaccination, was not associated with NDD, as defined in our study. However, the combination of preterm birth and vaccination was associated with a synergistic 7- to 14-fold increased odds of NDD [170].

The strength and consistency of the findings (e.g., in terms of links between diagnoses and medication use) suggest the possibility that some aspect of the current vaccination schedule, especially in the wake of preterm birth, which is known to be associated with maternal liver dysfunction [171,172], could be contributing to increases in autism and related chronic diseases in children. Follow-up studies are urged to confirm or refute these findings.

The present hypothesis suggests that *liver dysfunction,* occurring post-natally in children, may be a common feature in children with regressive autism. Earlier reports have linked autism and liver dysfunction [173,174]. A recent study assessed metabolic disorders (obesity, diabetes, hypertension, hyperlipidemia, nonalcoholic fatty liver disease [NAFLD] and nonalcoholic steatohepatitis [NASH]) in 48,762 individuals with ASD and 243,810 matched controls, identified through the Military Health System database [175]. Children with ASD were significantly more likely than controls to be diagnosed with NAFLD/NASH (OR 2.74; 95% CI: 2.06, 3.65) as well as obesity (8.2% vs. 4.7%; OR 1.85; 95% CI: 1.78, 1.92), Type 2 diabetes, hypertension and hyperlipidemia, and tended to have more severe disease than controls.

## 13. Could Other Risk Factors Be Involved?

Although the combined factors of preterm birth and vaccination were associated with up to 14-fold increased odds of NDD in our pilot study [169,170], these factors do not fully explain the growing nationwide epidemic of ASD. The model presented here suggests that other factors may be contributing to autism and associated outcomes, similarly via liver dysfunction and/or hypervitaminosis A in infants and pregnant women.

One factor could be vitamin A supplementation itself. A randomized trial, carried out at the Bandim Health Project, Guinea-Bissau, determined the impact of vitamin A supplementation (VAS) versus the oral polio vaccine (OPV) on low birthweight (LBW) male infants. VAS was associated with a significantly increased mortality rate compared to the OPV group (Hazard Ratio 9.91; 95% CI: 1.23, 80) and led to an abrupt halt to the trial [176]. The VAS group also had poorer weight gain and smaller lower arm circumference. A prospective birth cohort study in Japan showed that supplemental vitamin A intake prior to and/or during pregnancy was significantly associated with disturbed behavior at three years of age, after adjustment for numerous factors (age, number of deliveries, infertility treatment, consumption of fast food, smoking status, maternal and paternal education and income, gestational age at birth, infant’s weight, height, head circumference and body circumference, and the State-Trait Anxiety Inventory at age three) [177].

Benn et al. have commented on the apparent enigma that vitamin A supplementation does not always reduce mortality [178], even though putative vitamin A deficiency is associated with increased mortality. Severe vitamin A deficiency is commonly defined as a serum retinol concentration <0.7 µmol/L [38,179]. However, as noted, a low serum retinol concentration does not necessarily indicate low overall vitamin A nutritional status; it may instead indicate vitamin A accumulation and toxicity in the liver due to impaired hepatic mobilization and secretion, and an overall state of vitamin A toxicity due to the spillage of unbound retinyl esters and retinoic acids into the circulation.

The World Health Organization recommends high-dose vitamin A supplementation for infants and young children in places where vitamin A deficiency is a public health problem, as indicated by a prevalence rate of night blindness of 1% or higher in children 24–59 months of age, or in situations where vitamin A deficiency (serum retinol ≤ 0.7 µmol/L) is 20% or higher in infants and children 6–59 months of age. However, the policy and practice of administering high-dose vitamin A (defined as 25,000 to 50,000 IU vitamin A, equivalent to 7.5 mg and 15 mg retinol equivalents, respectively) to reduce child mortality has been strongly challenged and has led to reports of acute toxicity [180], ranging from increased intracranial pressure, bone demineralization and growth restriction, to mental retardation and death [181,182]. In a review of 21 studies evaluating the effect of vitamin A supplementation in community settings on all-cause mortality, 12 also reported cause-specific mortality for diarrhea and pneumonia and six reported measles-specific mortality. Combined results from six studies showed that neonatal vitamin A supplementation reduced all-cause mortality by 12% (Relative risk (RR) 0.88; 95% confidence interval (CI) 0.79–0.98) but had no effect on reducing all-cause mortality in infants 1–6 months of age (RR 1.05; 95% CI: 0.88, 1.26). Pooled results showed that vitamin A supplementation reduced all-cause mortality by 25% (RR 0.75; 95% CI: 0.64, 0.88) in children 6–59 months of age, but had no effect on measles (RR 0.71, 95% CI: 0.43, 1.16), meningitis (RR 0.73, 95% CI: 0.22, 2.48) and on pneumonia-specific mortality (RR 0.94, 95% CI: 0.67, 1.30) [183].

Reduced vitamin A concentration is believed to increase the risk of blindness in children with measles. However, a review of clinical trials since 1950 found that no trial of vitamin A supplementation had been published for the prevention of blindness or other ocular morbidities in children [184]. A systematic review of the safety and efficacy of high-dose vitamin A supplementation for treating severe acute malnutrition also found, in mainly well-nourished children, that high-dose vitamin A supplementation was associated with respiratory infection, diarrhea and growth restriction [185]. Interactions between vitamin A supplements and vaccines have also been reported in which the effects were harmful. For instance, children who received vitamin A with diphtheria, tetanus and pertussis (DTP) vaccine were at greater risk of death than children who received vitamin A alone or received nothing [186]. Given these data, a reassessment is warranted of the concept of vitamin A deficiency, the role of vitamin A in ophthalmic disorders and overall health, and the public practice of vitamin A supplementation.

A related factor is vitamin A-fortified infant foods and formula. A British report on vitamins A- and E-fortified infant foods and formulas concluded that the total daily intake of both vitamins exceeded recommendations of the U.K. Department of Health [187]. Consistent with the present model, a survey of the parents of 861 children with autism and 123 control children found that use of infant formula versus exclusive breastfeeding was associated with a four-fold increase in the odds of autism when all cases were considered (OR 4.41; 95% CI: 1.24, 15.7). However, when the analysis was restricted to cases in whom regression had occurred during development, the odds of autism rose to 12.96 (95% CI: 1.27, 132.0) [188].

Vitamin A-containing dairy foods such as milk and cheese are major constituents of the ‘Western diet’ (WD). Average intakes of cheese increased from 11.4 pounds/person per year in 1970 to 33.5 pounds/person in 2012 [189]. The suggestion that the Western diet itself could be contributing to the epidemic of childhood autism was supported by a study in which WD-treated female mice exhibited social avoidance, decreased body-body contacts, hyper-locomotion, and deficits in memory, compared to controls. Noting that the induced behaviors were reminiscent of the symptoms of autism, the authors suggested that the WD could exacerbate ASD [190].

## 14. Nutritional Implications of the Model

The data reviewed above suggest that restriction of vitamin A-containing foods could ameliorate the signs and symptoms of autism and related chronic illnesses. Consistent with this hypothesis, the case was reported of a female child who regressed into autism and responded poorly to conventional treatments. After starting a gluten-free, casein-free (GFCF, i.e., low dairy) diet, her autistic and medical symptoms markedly improved, but at the onset of puberty she developed seizures that could not be fully controlled with drugs. When placed again on a GFCF regimen while continuing on anticonvulsants, she experienced significantly reduced seizure activity. Of special interest, this diet is based on medium-chain triglycerides rather than on vitamin A-containing butter and cream as the primary source of fat. It was on this diet that the child’s morbid obesity resolved and both cognitive and behavioral features improved. Over the course of several years, her score on the Childhood Autism Rating Scale fell from 49 to 17; she became non-autistic and seizure-free, and her intelligence quotient increased 70 points [191].

A recent review of research on the GFCF diet for autism concluded that, with few exceptions, there were no significant differences in core symptoms of autism between treatment and control groups in terms of standardized measures [192]. However, some GFCF diets include vitamin A [187]. It is therefore difficult to determine what aspect of the diet could be having a salutary effect; furthermore, sensitivity to gluten-containing foods could be a consequence rather than a cause of autism. Tests of the present hypothesis involving nutritional interventions should therefore focus on determining the impact of removing preformed vitamin A itself from experimental diets.

In a randomized, single-blind, controlled nutritional and dietary trial in persons with ASD [193], involving 67 U.S. children and adults ages 3–58 years and 50 non-sibling neurotypical controls of similar age and gender, treatment began with a special vitamin/mineral supplement that included vitamin A. Later treatments included a GFCF diet added sequentially over a 12-month period. A significant improvement in nonverbal intellectual ability occurred in the treatment group as well as significant improvement in autism symptoms and developmental age. Results also showed a significant increase in serum vitamin A concentrations, suggesting improved liver function and vitamin A metabolism and thus the elimination of vitamin A toxicity.

In further support of the proposed model, the physical and mental health of a patient with latherosterolosis was reported to improve significantly after receiving a liver transplant [194]. Latherosterolosis, a rare metabolic disease associated with liver failure and autism, is attributed to a defect in cholesterol synthesis characterized by multiple congenital anomalies, mental retardation, and progressive liver disease. The case report is reminiscent of CRS and describes the only living patient with the condition. When diagnosed at age 2, she exhibited autistic behavior, was unable to walk without assistance, and her vision was impaired by cataracts. At age 7, she developed end-stage liver disease and received a liver transplant. Only one year later she was walking unaided, interacting with her caregivers, and her cholesterol synthesis was normal. By age 12, mental deterioration had ceased and she had a normal everyday life, albeit with limitations. While the authors speculated that her central nervous system had benefited from the normalization of cholesterol levels after liver transplantation, the present model suggests that the beneficial outcomes may have resulted from the normalization of vitamin A metabolism and function. In summary, just as measures taken by pregnant women infected with rubella virus to lower retinoid exposure and expression might lower the risk of fetal damage and CRS, a low vitamin A diet could be similarly beneficial for children with ASD.

## 15. Environmental Factors—Herbicides

Glyphosate, sold under the trade name ‘Roundup’, is a widely used herbicide, especially on genetically modified crops (including soy, corn, canola and sugar beets) that are resistant to its toxic effects. Glyphosate kills weeds and other plants by inhibiting the shikimate metabolic pathway, which is vital for plants and some microorganisms. Since glyphosate has been linked to a number of diseases, including autism, through prenatal exposure, calls for independent research have been issued [195]. In one study, an offspring’s risk of ASD with intellectual disability was increased following prenatal exposure to ambient glyphosate within 2000 m of their mother’s residence during pregnancy, compared with the offspring of women from the same agricultural region without such exposure (OR 1.33; 95% CI: 1.05, 1.69) [196]. It has been suggested that glyphosate, acting as a non-coding amino acid analogue of glycine, could be erroneously incorporated into protein synthesis in place of glycine, producing a defective product that resists proteolysis and leads to autism and other autoimmune diseases through the mechanism of molecular mimicry [197].

Using the zebrafish vertebrate model system, early exposure to glyphosate was found to cause morphological abnormalities including cephalic and eye reductions and a loss of delineated brain ventricles, along with decreased gene expression in the eye, forebrain and midbrain. Alterations in retinoic acid expression were also detected in the retina after 24-h exposure, suggesting that glyphosate is developmentally toxic to the forebrain and midbrain [198]. This evidence of teratogenicity and its impact on retinoic acid expression suggests that exposure to glyphosate could contribute to ASD via retinoid toxicity.

## 16. Discussion and Conclusions

A novel synthesis of the literature has been proposed on the mechanisms and relationship between rubella virus infection, the congenital rubella syndrome (CRS) and the link to autism, in which autism is both an aspect of the CRS and the mainly post-natal phenomenon of regressive autism. Some elements of the suggested pathogenesis (e.g., the metabolism of retinal to retinoic acid, the affinity of rubella virus for specific tissues) could be tested using induced pluripotent stem cells and/or iPSC-derived embryonic germ layer cells, serving as a model for the early steps of human development.

Specifically, it is proposed that:The signs and symptoms of rubella may be due to virus-induced alterations in vitamin A metabolism and its accumulation in the liver, leading to mild hepatic inflammation and dysfunction and to the spillage of stored vitamin A compounds into the circulation in correspondingly low concentrations and hence mild toxicity.RS and associated autism due to rubella infection in the early weeks of pregnancy may similarly result from maternal liver dysfunction and exposure of the fetus to excess endogenous vitamin A, resulting in embryopathy and long-term metabolic and neurodevelopmental disorders.Increasing rates of regressive autism result primarily from post-natal factors to which almost all children are exposed, causing liver dysfunction and both acute and chronic forms of hypervitaminosis A.These post-natal factors may include: multiple vaccinations and their ingredients administered together or in close succession, especially when combined with the physiological impact of preterm birth; the intake of high vitamin A-containing and vitamin A-fortified foods such as dairy products, infant formula, and vitamin A supplements; and exposure to certain ubiquitous herbicides that may influence retinoid metabolism. Further studies are awaited on the overall safety and health outcomes of the current vaccination schedule, with a view to optimizing the beneficial impact of vaccines on children’s health.An early diagnosis of ASD and CRS-like features in infants may be due to maternal exposures of a liver-damaging nature in the early weeks of pregnancy.Young children with liver-related conditions that include preterm birth and metabolic disorders, as well as genetic factors associated with alterations in vitamin A metabolism, may be especially susceptible to autism and related neurodevelopmental disorders.

The model could be tested by comparing children with autism and controls in terms of liver function, vitamin A enzyme activity, and retinoid concentration profiles (retinol, retinyl esters, percent retinyl esters, and retinoic acid). Children with autism would be expected to have significantly reduced vitamin A metabolic and catabolic enzyme activity, increased retinoic acid concentrations and receptor expression, low or normal concentrations of retinol and its transporter, RBP, and an increased percentage of serum retinyl esters as a fraction of total vitamin A (retinyl esters plus retinol). Subject to testing, novel dietary and clinical measures could be investigated for the treatment of ASD and associated allergic and metabolic disorders.

## Figures and Tables

**Figure 1 ijerph-16-03543-f001:**
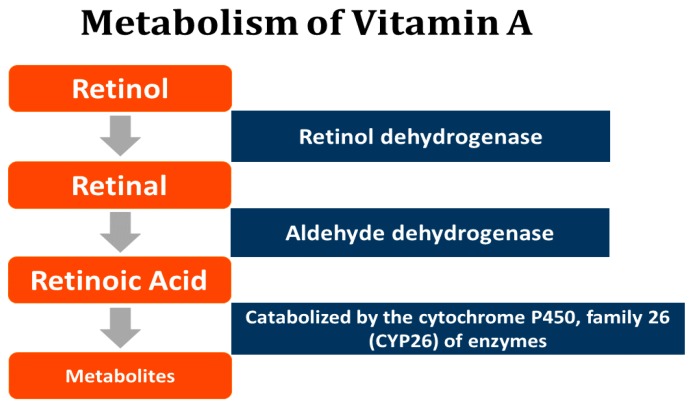
Metabolism of Vitamin A.

**Figure 2 ijerph-16-03543-f002:**
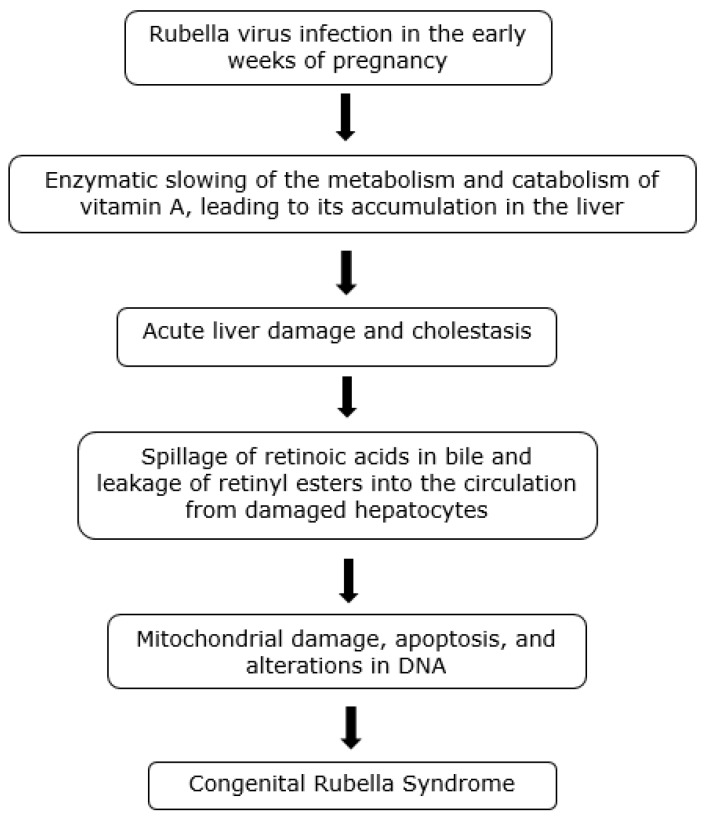
Authors’ proposed pathogenesis of the Congenital Rubella Syndrome.

**Table 1 ijerph-16-03543-t001:** Similarities between rubella symptoms and complications and acute hypervitaminosis A.

Symptoms/Signs	Rubella	Hypervitaminosis A
Mild fever	+	+
Arthralgia	+	+
Polyarthritis	+	+
Myalgia	+	+
Headache	+	+
Flu-like symptoms	+	+
Conjunctivitis	+	+
Lymphadenopathy	+	+
Maculopapular rash	+	+
Pruritus	+	+
Fatigue	+	+
Anorexia	+	+
Slight desquamation	+	+
Encephalitis/Encephalopathy	+	+
Splenomegaly	+	+
Miscarriage	+	+
Thrombocytopenic purpura	+	+
Guillain–Barré syndrome	+	+

References. Rubella: [7,8,11]; Hypervitaminosis A: [43,45,57,58,59,60,61,62,63,64,65,66,67].

**Table 2 ijerph-16-03543-t002:** Features of the congenital rubella syndrome resemble those of hypervitaminosis A-associated embryopathy and other effects.

Features	Congenital Rubella Syndrome	Hypervitaminosis A-Associated Effects
Cataract	+	+
Microphthalmia	+	+
Retinopathy	+	+
Bulging fontanelle	+	+
Intrauterine growth restriction	+	+
Seizures	+	+
Hearing defects	+	+
Heart defects, including Patent Ductus Arteriosus	+	+
Microcephaly	+	+
Cognitive deficits, behavioral and speech disorders	+	+
Hepatitis/jaundice	+	+
Hepatosplenomegaly	+	+
Bone lesions/osteoporosis	+	+
Type 1 diabetes mellitus	+	+
Thyroiditis	+	+
Encephalitis/Encephalopathy	+	+

References. Rubella: [7,11,31,74,89]; Hypervitaminosis A: [43,45,57,58,59,60,61,85,87,90,91,92,93].

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
