# Peer review of "Rubella Virus Infection, the Congenital Rubella Syndrome, and the Link to Autism"

_ijerph, 2019, doi:10.3390/ijerph16193543_

Round 1

Reviewer 1 Report

The manuscript submitted by Mawson & Croft is based on the hypothesis that the symptoms of congenital rubella syndrome are caused by exposure of the fetus to retinoids released in excess from maternal liver. This is a novel aspect of the literature on congenital rubella syndrome, rubella virus-associated autism and hypervitaminosis A.

Line 22 (page 2)– what is the time frame for vitamin A teratogenicity? Does it mirror the one for rubella virus, which is only teratogenic if infection occurs during the first 16 weeks of pregnancy as stated in line 14. Line 8 (page 4) – please specify “other” for more clarity. The headings and some paragraphs should be revised – “5. Rubella infection and CRS”, “8. Congenital Rubella Syndrome (CRS)”, and “9. Patholgy of CRS” could be combined – the abbreviation CRS should be used consistently. Heading Nr. 15 could refer to environmental factors associated with retinoid toxicity instead of Herbicides. Line 36 (page 18) - Retinoic acid and rubella virus were already tested on iPSCs – this could strengthen the manuscript and provide a new aspect. Just to name two as an example, the review by Gregg Duester, Cell, 2008, describes retinoic acid signaling during embryogenesis; L. J. Gudas, Semin Cell Dev Biol, 2013, summarizes retinoid-dependent stem cell differentiation. In this context, iPSCs could be indeed a valuable model to analyze the hypotheses made by Mawson & Croft. Line 38 to 43 – in agreement with line 47 (page 18) and line 1 to 8 (page 19) statements should be made as may be/could be/are hypothesized as. All points were introduced as “it is proposed that”. However, in its current format, the points 1 to 6 are not consistent and it should be clear, that all points are hypothesis-based.

One additional comment to the response to the previous submission:

In response to “Are there symptoms that might have another etiology than dysregulated vitamin A metabolism?” the authors stated that they hypothesize, that all signs and symptoms of CRS are due to the pleiotropic effects of retinoic acid. They could discuss, that there could be additional mechanisms involved in CRS besides retinoic acid metabolism.

Reviewer 2 Report

The revised version of "Rubella virus infection, the Congenital Rubella Syndrome, and the link to Autism" corrects the typographical errors that I previously detected. I continue to think that the review provides insight and a testable hypothesis, and is thus appropriate for publication.

Author Response

The revised version of "Rubella virus infection, the Congenital Rubella Syndrome, and the link to Autism" corrects the typographical errors that I previously detected. I continue to think that the review provides insight and a testable hypothesis, and is thus appropriate for publication.

Response: We thank the reviewer for the positive and encouraging comments.

Reviewer 3 Report

I wish to thank you for the opportunity to review this manuscript in which the authors propose that CRS is caused by maternal liver damage and consequent hypervitaminosis A. The authors also propose that the hypervitaminosis A is the cause of autism seen in some patients with CRS. Autism is a condition of multifactorial inheritance and many genetic and environmental factors have been demonstrated to play a role in its pathogenesis, and there is evidence of different congenital infections and pharmacological teratogens (misoprostol, valproic acid) involved in its etiology. Therefore, examining potential mechanistic pathways in the etiology of ASD is interesting. However, I cannot recommend the publication.

The manuscript is composed by an extensive review but don’t acknowledge its limitations, and do not include publications that are not consistent with their hypothesis. Moreover, their hypothesis is based on loose connections between different lines of evidence.

Retinoid embryopathy is known mostly secondary to synthetic retinoids (isotretinoin). On the other side, there are few human case reports of excess intake of vitamin A during pregnancy and defective outcomes. The retinoid embryopathy (RE) is not similar to CRS. Although they can share some common defects (microcephaly, cardiopathy) the differences are much more striking (e.g. external ear defects seen in RE is not present in CRS; and cataracts seen in CRS are not a hallmark of RE). Moreover, autism is not associated to RE as the authors suggest.  Moreover, there is some evidence that deficiency of vit A can be related to ASD, and animal models have shown that the lack of RA in developing brain is associated to ASD symptoms in rodents. In this sense, the recommendations the authors propose in restricting vit A to avoid ASD or CRS seems not only not scientifically sound, but also with potential iatrogenic consequences.

It is not clear why the authors try to associate the increase of autism in recent decades to CRS. If they theory was true, we would expect a temporal decrease in autism after the vaccination was implemented. Even the “regressive autism” couldn’t be related to CRS, since vaccination and rubella eradication is implemented since at least 3 decades even in developing countries as Latin America. The authors cite Hutton (2015) saying that CRS still occur in vaccinated populations – although this is true, the proportion of cases of rubella around the world have diminished significantly.

It is very worrying that the authors bring again the hypothesis that vaccines (especially MMR) might be related to ASD (pg 20, conclusion 4) The authors were not able to include any scientific reference for this affirmation (because there are none) and failed to review the extensive evidence against this spurious association.

Author Response

I wish to thank you for the opportunity to review this manuscript in which the authors propose that CRS is caused by maternal liver damage and consequent hypervitaminosis A. The authors also propose that the hypervitaminosis A is the cause of autism seen in some patients with CRS. Autism is a condition of multifactorial inheritance and many genetic and environmental factors have been demonstrated to play a role in its pathogenesis, and there is evidence of different congenital infections and pharmacological teratogens (misoprostol, valproic acid) involved in its etiology. Therefore, examining potential mechanistic pathways in the etiology of ASD is interesting. However, I cannot recommend the publication.

The manuscript is composed by an extensive review but don’t acknowledge its limitations, and do not include publications that are not consistent with their hypothesis. Moreover, their hypothesis is based on loose connections between different lines of evidence.

Response: We thank the review for the critical comments. We have outlined a testable and comprehensive hypothesis of the pathogenesis of rubella infection, the CRS, and the link to regressive autism. Once the hypothesis is tested, its potential strengths and limitations will become manifest.

Reviewer: Retinoid embryopathy is known mostly secondary to synthetic retinoids (isotretinoin). On the other side, there are few human case reports of excess intake of vitamin A during pregnancy and defective outcomes. The retinoid embryopathy (RE) is not similar to CRS. Although they can share some common defects (microcephaly, cardiopathy) the differences are much more striking (e.g. external ear defects seen in RE is not present in CRS; and cataracts seen in CRS are not a hallmark of RE).

Response: The literature reviewed in our paper suggests strong parallels between the features of CRS and vitamin A teratogenicity.

Reviewer: Moreover, autism is not associated to RE as the authors suggest.  Moreover, there is some evidence that deficiency of vit A can be related to ASD, and animal models have shown that the lack of RA in developing brain is associated to ASD symptoms in rodents. In this sense, the recommendations the authors propose in restricting vit A to avoid ASD or CRS seems not only not scientifically sound, but also with potential iatrogenic consequences.

Response: The vitamin A toxicity hypothesis as proposed in our paper has not yet been tested. As we point out in the paper, low serum retinol concentrations are often assumed to indicate deficiency but may in fact be due to impaired liver secretion of retinol-binding protein and to the presence of increased retinyl ester concentrations, i.e. retinoid toxicity. Our review of the literature on regressive autism suggests that current dietary and even surgical interventions could be effective in part by lowering vitamin A concentrations.

Reviewer:  It is not clear why the authors try to associate the increase of autism in recent decades to CRS. If they theory was true, we would expect a temporal decrease in autism after the vaccination was implemented. Even the “regressive autism” couldn’t be related to CRS, since vaccination and rubella eradication is implemented since at least 3 decades even in developing countries as Latin America. The authors cite Hutton (2015) saying that CRS still occur in vaccinated populations – although this is true, the proportion of cases of rubella around the world have diminished significantly.

Response: Rubella virus infection and CRS have indeed declined worldwide due to vaccination. However, our hypothesis is that the proposed mechanism of CRS – virus-associated liver dysfunction and exposure to excess endogenous vitamin A (retinyl esters and retinoid acid) – could be occurring independently of rubella infection itself, due to exposure to the other factors proposed in the paper.

Reviewer: It is very worrying that the authors bring again the hypothesis that vaccines (especially MMR) might be related to ASD (pg 20, conclusion 4) The authors were not able to include any scientific reference for this affirmation (because there are none) and failed to review the extensive evidence against this spurious association.

Response: The epidemiological evidence against the alleged association between vaccination and autism almost exclusively involves comparisons in which the groups (cases and controls) are vaccinated to some degree. There are no studies (actually very few) comparing autism rates in vaccinated and completely unvaccinated children. Our view is that further research is needed on the hypothesis of vaccination-related autism.

Round 2

Reviewer 3 Report

Thanks for the authors and I now agree with their responses.